# Identification of an Immune-Related Gene Signature for Prognostic Prediction in Glioblastoma: Insights from Integrated Bulk and Single-Cell RNA Sequencing

**DOI:** 10.3390/cancers17111799

**Published:** 2025-05-28

**Authors:** Jianan Chen, Qiong Wu, Anders E. Berglund, Robert J. Macaulay, James J. Mulé, Arnold B. Etame

**Affiliations:** 1Department of Neuro-Oncology, H. Lee Moffitt Cancer Center and Research Institute, Tampa, FL 33612, USA; jianan.chen@moffitt.org (J.C.); qiong.wu@moffitt.org (Q.W.); 2Department of Quantitative Health Sciences, Division of Computational Biology, Mayo Clinic, 4500 San Pablo Road South, Jacksonville, FL 32224, USA; berglund.anders@mayo.edu; 3Departments of Anatomic Pathology, H. Lee Moffitt Cancer Center and Research Institute, 12902 Magnolia Drive, Tampa, FL 33612, USA; robert.macaulay@moffitt.org; 4Department of Immunology, H. Lee Moffitt Cancer Center and Research Institute, 12902 Magnolia Drive, Tampa, FL 33612, USA; james.mule@moffitt.org

**Keywords:** glioblastoma, immune signature, tumor microenvironment, single-cell RNA-seq, macrophages, proliferation TAM, angiogenesis, prognosis

## Abstract

Glioblastoma is the most aggressive type of brain cancer and is very difficult to treat. One reason for this is that the tumor is well-surrounded by special immune cells called macrophages, which normally help fight infections but are instead helping the tumor grow. In this study, we identified five genes that are often active in these harmful macrophages. We found that patients with higher levels of these genes had worse survival outcomes. Our study showed that these genes are closely linked to tumor growth and the suppression of the body’s immune response. We also discovered that patients with high levels of these genes may respond better to certain cancer drugs. These findings could help doctors better predict how aggressive cancer will be and choose treatments that are more likely to work.

## 1. Introduction

Glioblastoma (GBM) is the most common and aggressive primary brain tumor in adults [1]. Despite intensive therapy, including surgical resection, radiotherapy, and chemotherapy, the median survival remains within two years [2,3]. Immunotherapies, such as immune checkpoint inhibitors (ICIs) and adoptive cell therapies, have shown promise in other cancers but have had limited impact on GBM [4]. The tumor’s highly immunosuppressive and heterogeneous microenvironment likely contributes to these disappointing outcomes, highlighting the need to better understand the underlying cellular and molecular landscape.

Within the immune ecosystem of GBM, tumor-associated macrophages (TAMs) account for up to 50% of total live cells in the whole tumor mass [5]. TAMs arise from both brain-resident microglia and peripheral monocyte-derived macrophages, exhibiting a remarkable capacity to switch between pro-inflammatory and immunosuppressive phenotypes [6]. These cells regulate various processes such as tumor growth, angiogenesis, and resistance to therapy. Moreover, a high TAM density correlates with a worse prognosis, highlighting the importance of characterizing TAM subpopulations and their specific functional roles in GBM progression [7,8].

Bulk RNA sequencing has revealed valuable biomarkers and molecular subtypes of GBM, but it cannot fully capture the cellular heterogeneity within the tumor. In contrast, single-cell RNA sequencing (scRNA-seq) allows for the precise analysis of individual cell populations, providing deeper insight into TAM diversity and other cellular components of the tumor microenvironment [9,10]. This single-cell approach is invaluable for identifying previously unrecognized molecular mechanisms that drive GBM progression and for uncovering potential therapeutic targets [11].

In this study, we integrated bulk RNA-seq and scRNA-seq data to identify an immune-related gene signature—*THEMIS2*, *SIGLEC9*, *CSTA*, *LILRB3*, and *MS4A6A*—that is strongly associated with poor survival in GBM. Pathway enrichment analyses revealed that these genes are involved in critical immune processes, including antigen presentation, cytokine signaling, and immune cell activation. A single-cell analysis demonstrated that these genes are highly expressed in “proliferation” TAMs, indicating their potential role in driving tumor angiogenesis and expansion. Furthermore, drug sensitivity analyses revealed distinct therapeutic vulnerabilities between high- and low-risk groups, suggesting that this Macrophage-Associated Prognostic Signature (MAPS) could also serve as a valuable tool for guiding personalized immunotherapy-based strategies for GBM patients.

Overall, our study sheds light on the immune landscape of GBM and identifies a gene signature with significant prognostic and therapeutic implications. By elucidating how proliferation TAMs contribute to tumor progression and revealing their associated therapeutic vulnerabilities, we emphasize the importance of developing targeted therapies for these cells. This research establishes a foundation for more precise immunotherapeutic interventions, aiming to improve clinical outcomes for GBM patients.

## 2. Materials and Methods

### 2.1. GBM Datasets and Preprocessing

The gene expression and clinical data of glioblastoma patients were obtained from the TCGA database (https://tcga-data.nci.nih.gov/ (accessed on 27 March 2025)) and the CGGA database (http://www.cgga.org.cn/ (accessed on 27 March 2025)). Specifically, the CGGA mRNAseq_325 dataset [12], consisting of 139 GBM patients and 24,326 genes, and the CGGA mRNAseq_693 dataset [13], including 249 GBM patients and 23,987 genes, were used. The TCGA dataset included 159 GBM patients with 60,660 genes. Additionally, single-cell RNA-sequencing data were sourced from the GSE131928 [14] dataset in the GEO database (https://www.ncbi.nlm.nih.gov/geo/ (accessed on 27 March 2025)). After quality control, 14,472 cells were retained for downstream analysis.

The somatic mutation data for GBM patients were retrieved from the TCGA database using the “TCGAbiolinks (version 2.27.0)” [15] R package. Masked somatic mutation data under the workflow type “Aliquot Ensemble Somatic Variant Merging and Masking” were downloaded and prepared for analysis. Mutation allele frequencies (MAFs) were calculated for each mutation. The tumor mutation burden (TMB) and mutation spectrum were further analyzed using the “maftools (version 2.13.0)” [16] R package. Somatic mutation data were formatted into Mutation Annotation Format, and oncoplots were generated to visualize the top 20 most frequently mutated genes and their mutation types. TMB values were calculated for each sample, assuming a sequencing capture size of 50 Mb.

### 2.2. Construction and Validation of an Immune-Related Gene Signature Risk Model

Survival analysis was independently conducted for each gene in the TCGA, CGGA mRNAseq_325, and CGGA mRNAseq_693 datasets using the ‘survival (version 3.8-3)’ and ‘survminer (version 0.5.0)’ R packages. Patients were stratified into high- and low-expression groups based on the median expression value of each gene, and the log-rank test was performed to identify survival-associated genes with *p*-values < 0.05. The intersection of these genes across the datasets resulted in 41 survival-associated genes. Using the STRING database (https://string-db.org/ (accessed on 27 March 2025)), we further analyzed the interactions of these genes and identified a subset of immune-related gene signatures that are significantly involved in immune pathways and processes. Subsequently, the interaction network of these immune-related genes was visualized using GeneMANIA (https://genemania.org/ (accessed on 27 March 2025)), incorporating data on co-expression, physical interactions, and shared protein domains to illustrate their functional relationships.

Ridge regression analysis was performed using the “glmnet (version 4.1-8)” [17] R package to construct the risk model. The response variable (y) was overall survival (OS) and survival status (Censor), modeled using a Cox proportional hazards framework. Gene expression levels for five genes (*CSTA*, *LILRB3*, *MS4A6A*, *SIGLEC9*, and *THEMIS2*) were extracted from the dataset and standardized. The optimal regularization parameter (lambda) was determined through tenfold cross-validation. The risk score for each patient was calculated as follows:(1)$$Risk\;score=\sum \left(\beta_{i}\cdot x_{i}\right)$$

The regression coefficients obtained from the ridge regression model were as follows: *CSTA*: 0.0276; *LILRB3*: 0.0635; *MS4A6A*: −0.0032; *SIGLEC9*: 0.0091; and *THEMIS2*: −0.0035. The presence of negative coefficients is due to the strong correlation among these genes. Ridge regression, which applies L2 regularization, redistributes weights to minimize multicollinearity effects, sometimes leading to sign changes in coefficients. However, all five genes are individually associated with worse survival in univariate Cox regression analysis, and the final risk score remains predictive of glioblastoma prognosis. The surv_cutpoint function from the “survminer” R package was used to identify the optimal cutoff point for stratifying patients into high-risk and low-risk groups based on their risk scores. A total of 65 patients were classified into the low-risk group, while 94 patients were classified into the high-risk group. The optimal cutoff point for risk scores was determined to be 0.102793786.

A heatmap was generated using the “pheatmap (version 1.0.12)” R package to visualize the expression of the gene signature (*CSTA*, *LILRB3*, *MS4A6A*, *SIGLEC9*, and *THEMIS2*) across risk groups. The ggsurvplot function from the “survminer” package was employed to visualize survival curves of the high and low-risk groups. Univariate Cox regression analyses were performed for the selected genes and the calculated risk score using the coxph function from the “survival (version 3.8-3)” package. Forest plots were created using the “forestplot (version 3.1.6)” R package to visualize the results. Bar plots were created using the “ggplot2 (version 3.5.2)” [18] package to visualize the distribution of risk scores across samples. Survival times and statuses (Alive/Dead) were visualized with scatter plots to provide a comprehensive view of patient outcomes.

### 2.3. Nomogram Construction and Evaluation

To provide an individualized prognostic prediction tool, a nomogram was constructed by integrating the risk score with clinical variables, including age, gender, and race. The nomogram was designed to estimate the 1-year, 2-year, and 3-year overall survival (OS) probabilities for each patient. The “rms (version 8.0-0)” R package was used to create the nomogram [19]. Calibration curves were then generated to assess the agreement between the predicted survival probabilities from the nomogram and the actual outcomes. The calibration curves for 1-year, 2-year, and 3-year OS were plotted using the “calibrate” function in the “rms” package, ensuring the reliability and accuracy of the predictions.

To evaluate the predictive performance of the nomogram, time-dependent receiver operating characteristic (ROC) curves were plotted using the “timeROC (version 0.4)” R package. The area under the curve (AUC) was calculated for the 1-year, 2-year, and 3-year OS to quantify the predictive ability of the nomogram.

### 2.4. Functional Enrichment and Pathway Analyses

The biological roles and pathways related to differentially expressed genes (DEGs) between high-risk and low-risk groups were explored through Gene Ontology (GO) [20], Kyoto Encyclopedia of Genes and Genomes (KEGG) [21], and Gene Set Enrichment Analysis (GSEA) [22]. DEGs were identified from RNA-sequencing data of the TCGA dataset, with criteria set at an adjusted *p*-value < 0.01 and |log2 fold change| > 1. We performed GO and KEGG analysis on the top 700 most significantly differentially expressed genes. GO and KEGG enrichment analyses were carried out using the “clusterProfiler (version 4.9.5)” [23] R package. GO analysis categorized DEGs into biological processes (BPs), cellular components (CCs), and molecular functions (MFs). KEGG analysis highlighted significantly enriched pathways. Visualization of results was achieved using bubble and bar plots generated with the “ggplot2” R package [18]. The Benjamini–Hochberg method adjusted *p*-values (*p*.adjust), with terms showing *p*.adjust < 0.05 considered significant. For GSEA, the most significantly enriched pathway was identified and analyzed. The Cytokine–Cytokine Receptor Interaction pathway was highlighted, with the enrichment score (ES) calculated for the ranked gene list. The normalized enrichment score (NES) and false discovery rate (FDR) were used to assess the pathway’s significance, and the GSEA results were visualized using an enrichment curve.

### 2.5. Tumor Microenvironment and Immune Checkpoint Analysis

We analyzed the tumor microenvironment characteristics and immune checkpoint expression in high-risk and low-risk groups using multiple computational methods. StromalScore, ImmuneScore, and ESTIMATEScore were calculated with the ESTIMATE algorithm [24], and the results were visualized using box plots generated by the “ggplot2” R package [18]. The expression levels of 14 immune checkpoints, including CD163, PDCD1, CD274, CTLA4, and LAG3, were extracted and compared between risk groups, with statistical significance assessed via *t*-tests. Correlation between the gene signature and immune checkpoint expression was calculated using Pearson’s correlation, with heatmaps generated via the “corrplot (version 0.95)” package to illustrate the relationships.

### 2.6. Mutation and Tumor Mutation Burden Analysis

Somatic mutation data were retrieved from TCGA to assess the mutation landscape of high- and low-risk groups. Waterfall plots were used to visualize the most frequently mutated genes and their mutation types. TMB was calculated for each patient, and its distribution was compared between the two risk groups to identify potential differences in genomic instability.

### 2.7. Single-Cell Transcriptome Analysis

Single-cell RNA-sequencing data were processed to characterize cell populations in GBM tumor samples. Low-quality cells with fewer than 200 genes, more than 7500 genes, or mitochondrial gene content exceeding 10% were excluded to ensure data quality. Processed expression matrices were used to create Seurat objects using the “Seurat (version 5.3.0)” [25] R package.

Following normalization with the “LogNormalize” method and scaling, highly variable genes (top 2000) were identified. Principal component analysis (PCA) was performed, and the ElbowPlot function was used to determine the optimal number of components for downstream analysis. Clustering was performed using the “FindNeighbors” and “FindClusters” functions with a resolution of 0.4, and Uniform Manifold Approximation and Projection (UMAP) was used to visualize the clustering results. Cell clusters were annotated based on canonical marker genes [14], including macrophages (*CD14*, *AIF1*, *FECR1G*, *FCGR3A*, *TYROBP*, and *CSF1R*) (Appendix A), T cells (*CD3D*, *CD3E*, *CD2*, and *CD3G*) (Appendix A), oligodendrocytes (*MBP*, *TF*, *PLP1*, *MAG*, *MOG*, and *CLDN11*) (Appendix A), and tumors (*EGFR*, *SOX4*, *BCAN*, *PDGFRA*, *IDH1*, *GFAP*, *SLC1A3*, *VIM*, and *SOX2*) (Appendix A).

Differentially expressed genes were identified between macrophages with high and low gene expression scores, which were stratified based on the median value of the Gene Score module. The Gene Score was computed using the AddModuleScore function in Seurat, representing the combined expression of the five selected genes (*SIGLEC9*, *CSTA*, *MS4A6A*, *THEMIS2*, and *LILRB3*) across different macrophage subsets. Cells were classified into high- and low-risk groups based on whether their Gene Score was above or below the median. Differential expression analysis was then conducted using the FindMarkers function in Seurat to identify genes that are significantly upregulated or downregulated between these groups. Pathway enrichment analysis was performed using KEGG and GO terms as previously described. UMAP and joint density plots revealed that specific genes (*SIGLEC9*, *CSTA*, *MS4A6A*, *THEMIS2*, and *LILRB3*) were highly expressed and co-expressed within distinct macrophage populations.

### 2.8. Macrophage Classification and Analysis

Macrophages were further extracted for downstream analysis and saved as a separate Seurat object. PCA was performed on the processed single-cell RNA-seq data, followed by the construction of a nearest-neighbor graph and clustering with a resolution of 0.4. UMAP was used to visualize the clustering results. This resulted in 10 macrophage subpopulations, which were then annotated based on known gene markers [26], identifying four major subtypes: Mg-TAM (*CX3CR1*, *P2RY12*, and *FCGR1A*), Monocyte (*CD52*, *VCAN*, and *FCN1*), Mo-TAM (*CD14*, *CD163*, and *TGFBI*), Dendritic Cell (*AREG*, *FCER1A*, and *CLEC10A*), and Prol-TAM (*STMN1*, *TYMS*, and *MKI67*).

To investigate the transitional relationships among these subtypes, pseudotime trajectory analysis was performed using “Slingshot (version 2.6.0)” [27] R package. UMAP embeddings and Seurat cluster identities were used as inputs, and the starting cluster was determined based on the pseudotime distribution shown in the analysis (Appendix A). This trajectory analysis revealed two additional transition-state subtypes—Pre-TAM and Trans-TAM (Appendix A).

Finally, we assessed the expression of the gene signature by calculating module scores for each cell within Seurat. These scores were then incorporated into the metadata for cluster-wise comparisons. Boxplots illustrating the module score distributions were generated for each subtype, and pairwise *t*-tests were performed to evaluate statistical significance which was visualized using *p*-value annotations.

### 2.9. Validation of Key Gene Expression Patterns Using Ivy GAP

Gene expression data from the Ivy Glioblastoma Atlas Project (Ivy GAP) [28] (https://glioblastoma.alleninstitute.org/ (accessed on 27 March 2025)) were analyzed to validate findings from single-cell RNA-seq. Genes with no detectable expression were filtered out, and the data were log2-transformed. The top 1000 most variable genes were selected for principal component analysis (PCA), with PC1 and PC2 used to capture the major sources of variation.

Expression levels of *CSTA*, *SIGLEC9*, *LILRB3*, and *MS4A6A* were overlaid onto PCA plots to assess their distribution across samples. Additionally, samples were grouped based on metadata, and group information was integrated into the PCA results. Visualization was performed using “ggplot2” package [18].

### 2.10. Drug Sensitivity Analysis

Drug sensitivity analysis was performed using the “OncoPredict (version 1.1.0)” [29] package in R. Gene expression data were used to predict the half-maximal inhibitory concentration (IC50) values of various drugs based on the Cancer Therapeutics Response Portal (CTRP) training dataset [30]. Batch correction and filtering of low-variance genes were applied to ensure robust predictions. Differences in predicted IC50 values between high- and low-risk groups were assessed using the Wilcoxon rank-sum test, and the most significant small-molecule drugs (based on *p*-values) were identified and visualized using “ggplot2” package [18].

### 2.11. Statistical Analysis

All statistical analyses were conducted using R software (version 4.4.1). Survival analyses were performed using the “survival” and “survminer” packages, with Cox proportional hazard models applied to assess the prognostic significance of the gene signature. Differential expression and pathway enrichment analyses were conducted using the “limma” [31] and “clusterProfiler” packages, respectively. Comparisons of variables between two groups were performed using the Wilcoxon rank-sum test. For all analyses, *p*-values < 0.05 were considered statistically significant.

## 3. Results

### 3.1. Identification of an Immune-Related Prognostic Gene Signature in GBM

Through the analysis of the gene expression and clinical data from three independent GBM datasets (TCGA, CGGA mRNAseq_693, and CGGA mRNAseq_325), 2422, 2468, and 2864 genes, respectively, were identified as being significantly associated with overall survival (*p* < 0.05). The intersection of these genes resulted in 41 survival-associated candidates, among which five immune-related genes—*THEMIS2*, *SIGLEC9*, *CSTA*, *LILRB3*, and *MS4A6A*—were selected as a prognostic signature due to their involvement in key immune pathways (Figure 1A). The interaction network of the five prognostic genes was constructed using GeneMANIA, revealing strong functional connections through co-expression, physical interactions, and shared pathways (Figure 1B). These genes are central to immune processes and interacting with key immune regulators like *TYROBP*, *LILRB2*, and *FCER1G*. Then, a risk model was established to stratify patients into high-risk (n = 94) and low-risk (n = 65) groups based on an optimal risk score cutoff of 0.1028. Heatmap visualization demonstrated distinct gene expression patterns, with higher expression levels in the high-risk group (Figure 1C). Further Cox regression analyses, performed using the TCGA dataset, revealed a significant association between the gene signature and patient outcomes: four out of the five genes were independently associated with survival (*p* < 0.05; hazard ratios ranging from 1.48 to 1.54), while MS4A6A showed a trend toward significance (*p* = 0.055, HR = 1.4). The combined risk model demonstrated a hazard ratio of 2.0 (95% CI: 1.39–2.89, *p* < 0.001) (Figure 1D). Additionally, the survival status distribution confirmed the poor prognosis of high-risk patients (Figure 1E).

### 3.2. Nomogram-Based Prognostic Prediction and Model Evaluation

A nomogram was developed using the TCGA dataset to predict the 1-year, 2-year, and 3-year overall survival probabilities by integrating risk scores with clinical variables such as age, gender, and race (Figure 2A). The calibration plots for the nomogram demonstrated a good agreement between the predicted and observed survival probabilities at 1 year, 2 years, and 3 years, indicating the model’s reliability (Figure 2B–D).

To evaluate the predictive performance of the risk model, time-dependent receiver operating characteristic (ROC) curves were plotted. The area under the curve (AUC) values for 1-year, 2-year, and 3-year survival were 0.69, 0.64, and 0.71, respectively, confirming the moderate predictive power of the model (Figure 2E). Furthermore, a Kaplan–Meier survival analysis showed a significant difference in OS between the high-risk and low-risk groups (*p* = 0.00015), with patients in the high-risk group exhibiting significantly worse survival outcomes (Figure 2F).

### 3.3. Functional Enrichment Analysis

To explore the biological significance of DEGs between the high- and low-risk groups, a volcano plot was generated to visualize the DEGs (Figure 3A) in the TCGA dataset. The top 700 DEGs were identified based on the criteria of |log2 fold change| > 1 and adjusted *p*-value < 0.01. Notably, immune-related genes such as *CCL2*, *FPR1*, and *SERPINB1* were significantly upregulated in the high-risk group. The GO enrichment analysis revealed that the DEGs were significantly enriched in immune-related biological processes, including leukocyte-mediated immunity, cytokine production regulation, and myeloid leukocyte differentiation (Figure 3B, BP panel). The cellular component analysis highlighted their localization in the secretory granule lumen and cytoplasmic vesicle lumen, while the molecular function analysis identified significant enrichment in immune receptor activity and cytokine receptor binding (Figure 3B, CC and MF panels). The KEGG pathway enrichment analysis further demonstrated that the DEGs were involved in critical immune pathways, including cytokine–cytokine receptor interaction, hematopoietic cell lineage, and phagosome formation (Figure 3C). Notably, the cytokine–cytokine receptor interaction pathway was prominently enriched, emphasizing its central role in the immune regulation of GBM. The GSEA confirmed the significant enrichment of the cytokine–cytokine receptor interaction pathway in the high-risk group, with a high normalized enrichment score (NES) and low false discovery rate (FDR) (Figure 3D).

### 3.4. Tumor Microenvironment and Immune Checkpoint Analysis

The analysis of the gene signature revealed strong positive correlations with multiple immune checkpoints, including *HAVCR2*, *IL10RA*, *CSF1R*, and *CD163*, suggesting these genes play a role in regulating the immunosuppressive tumor microenvironment (Figure 4A). Consistent with this, the StromalScore, ImmuneScore, and ESTIMATEScore calculated using the ESTIMATE algorithm were significantly higher in the high-risk group compared to the low-risk group, indicating a more complex and immunosuppressive tumor microenvironment in high-risk patients (Figure 4B, *p* < 0.0001). A further detailed analysis of immune checkpoint expression between risk groups showed that critical immune checkpoint molecules such as *CD163*, *CD274*, *CSF1R*, *HAVCR2*, *IL10*, *IL10RA*, *PDCD1*, *PDCD1LG2*, *SIGLEC15*, and *TREM2* were significantly upregulated in the high-risk group (Figure 4C, all *p* < 0.01). These findings collectively suggest that the high-risk group is characterized by an immunosuppressive microenvironment driven by both stromal and immune components, which may serve as potential therapeutic targets for immune checkpoint blockade therapy.

### 3.5. Mutation Landscape and Tumor Mutation Burden

To compare the genomic alterations between the high- and low-risk groups, we analyzed the mutation landscape and tumor mutation burden of each group using the TCGA dataset. In the high-risk group (Figure 5A,B), 70 out of 81 samples (86.42%) displayed somatic mutations, with *PTEN*, *TP53*, and *TTN* being the most frequently mutated genes (46%, 31%, and 27%, respectively). Missense mutations were the most common type of alteration, followed by frameshift deletion and frameshift insertions. The median TMB in the high-risk group was 0.82 mutations per megabase (mutations/MB), indicating relatively low genomic instability. In the low-risk group (Figure 5C,D), 49 out of 53 samples (92.45%) exhibited somatic mutations, with *TP53*, *EGFR*, and *TTN* being the most frequently mutated genes (38%, 32%, and 26%, respectively). Similar to the high-risk group, missense mutations were predominant. However, the low-risk group showed a higher median TMB of 1.04 mutations/MB, suggesting slightly greater genomic instability compared to the high-risk group.

### 3.6. Single-Cell and Functional Analysis

To further investigate the cellular origin of the gene signature, we performed single-cell RNA-seq analysis using the GSM3828673 dataset. Based on previously described cell markers, cells were classified into four major populations: macrophages, malignant cells, T cells, and oligodendrocytes (Figure 6A). *THEMIS2*, *SIGLEC9*, *CSTA*, *LILRB3*, and *MS4A6A* were found to be highly expressed solely in macrophages, suggesting their significant role in this cell population (Figure 6B).

Furthermore, we extracted macrophages and conducted a deeper analysis. A UMAP visualization of macrophages stratified by the gene signature score revealed distinct high-expression and low-expression groups (Figure 6C). A differential expression analysis of macrophages with a high and low expression of the gene signature identified significant DEGs, which were subsequently used for a functional enrichment analysis. The GO enrichment analysis further revealed their involvement in processes like the positive regulation of leukocyte activation, cellular components like endocytic vesicles, and molecular functions such as MHC class II protein complex binding (Figure 6D). The KEGG pathways showed that these genes are involved in critical immune-related pathways, including phagosome formation, antigen processing and presentation, and hematopoietic cell lineage (Figure 6E). In addition, the GSEA revealed a significant association between the identified DEGs and the systemic lupus erythematosus (SLE) pathway, as indicated by a high enrichment score and statistical significance (Figure 6F).

### 3.7. Functional Analysis of Gene Signature in Macrophage Subtypes

To further investigate the functionality of the gene signature, macrophages were extracted for downstream analysis and clustered into subpopulations. A pseudotime trajectory analysis using “Slingshot” R package revealed the transitional relationships among macrophage subtypes (Figure 7A). Based on UMAP embeddings and known cell markers (e.g., *CX3CR1*, *CD52*, and *CD14*), macrophages were classified into seven subtypes: Mg-TAM, Mo-TAM, Prol-TAM, Pre-TAM, Trans-TAM, Monocyte, and Dendritic Cell (Figure 7B).

The module scores for the gene signature were calculated for each cell and compared across subtypes. The scores were significantly higher in Mg-TAM, Mo-TAM, and Prol-TAM. Specifically, Mg-TAMs are associated with immune surveillance and maintaining homeostasis in the central nervous system; Mo-TAMs are involved in inflammatory responses, regulating angiogenesis, and supporting tumor cell survival; and Prol-TAMs, which are associated with tumor proliferation, are involved in metabolic and angiogenic processes (Figure 7C).

Validation using the Ivy Glioblastoma Atlas Project further corroborated these findings. PCA analysis revealed a high expression of the gene signature components (*CSTA*, *SIGLEC9*, *LILRB3*, and *MS4A6A*) in the CT-HBV, CT-MVP, CT-PAN, and CT-PNZ regions (Figure 7D–H). These regions correspond to distinct glioblastoma microenvironmental niches: CT-HBV is characterized by hyperplastic blood vessels supporting enhanced tumor angiogenesis and vascularization; CT-MVP involves clusters of proliferative endothelial cells driving microvascular proliferation and metabolic support; CT-PAN represents pseudopalisading necrosis, with tumor cells exhibiting aggressive and invasive behavior under severe hypoxia; and CT-PNZ refers to the perinecrotic zone, where tumor cells adapt to hypoxic conditions and exhibit stem-like properties. *THEMIS2* was not included in the Ivy database, and, thus, no corresponding figure is provided.

### 3.8. Drug Sensitivity Analysis

The drug sensitivity analysis revealed significant differences in IC50 values between high-risk and low-risk groups, which were defined based on risk scores derived from a Ridge regression analysis using the TCGA dataset, highlighting distinct therapeutic vulnerabilities. The high-risk group demonstrated an increased sensitivity to BMS-536924, an IGF-1R inhibitor that targets tumor growth and survival pathways [32]; SCH772984, an ERK1/2 inhibitor that disrupts cell proliferation and progression via the MAPK pathway [33]; and Selumetinib, a MEK1/2 inhibitor that blocks a critical signaling cascade involved in tumor cell survival [34]. These drugs, particularly BMS-536924, also play roles in inhibiting angiogenesis, suggesting that high-risk patients may exhibit a heightened sensitivity to anti-angiogenic therapies [35]. Conversely, the low-risk group exhibited a higher sensitivity to GSK1904529A, another IGF-1R inhibitor that disrupts tumor growth and survival pathways [36]; Tozasertib, an Aurora kinase inhibitor that interferes with mitosis and induces cell cycle arrest [37]; Daporinad, a NAMPT inhibitor that targets tumor metabolism by depleting NAD+ levels [38]; PCI-34051, a selective HDAC8 inhibitor promoting apoptosis in cancer cells [39]; Vincristine, a microtubule-disrupting agent that inhibits cell division [40]; and Sepantronium bromide, an apoptosis-inducing agent targeting survivin, a key anti-apoptotic protein [41] (Figure 8).

## 4. Discussion

Glioblastoma remains the most aggressive primary brain tumor, and is often enriched with tumor-associated macrophages, which, in high-grade tumors, can constitute up to 30–50% of the total live cells [42,43]. Our study identified a novel five-gene, immune-related risk signature—*THEMIS2*, *SIGLEC9*, *CSTA*, *LILRB3*, and *MS4A6A*—that stratifies patients into high- and low-risk groups with distinct survival outcomes and a profound difference in their TAM-dominated microenvironments. This Macrophage-Associated Prognostic Signature (MAPS) could also serve as a valuable tool for guiding personalized immunotherapy-based strategies for GBM patients.

Interestingly, this macrophage-centric signature exhibited unexpected connections to autoimmune dysregulation. Despite the seemingly opposing immune landscapes of SLE (hyperactive immunity) [44] and GBM (immunosuppression), the GSEA revealed the significant enrichment of the SLE-related pathway in macrophages expressing this gene signature (Figure 6F). This finding aligns with emerging evidence that M2-like macrophages may contribute to pathogenic inflammation in both disease contexts: while promoting angiogenesis and immune evasion in GBM [45], analogous subsets exacerbate tissue damage in SLE through impaired immune complex clearance and lupus nephritis progression [46]. This observation suggests a potential convergence of immune regulatory programs, and highlights the plasticity of macrophage biology and raises the possibility that targeting such shared mechanisms may offer cross-disease therapeutic value. Further functional studies are warranted in order to validate this connection.

The high-risk GBM patients’ group showed a significantly elevated StromalScore, ImmuneScore, and ESTIMATEScore, and also overexpressed immune checkpoint molecules such as *PDCD1*, *HAVCR2*, and *CSF1R*, pointing to an environment shaped by immunosuppressive signals [47]. A functional enrichment analysis revealed that genes distinguishing between high- from low-risk groups were involved in leukocyte activation, cytokine–cytokine receptor interactions, and phagosome formation. These findings align with our single-cell RNA sequencing results, which placed the five-gene signature predominantly in TAMs, including subtypes associated with angiogenesis and proliferation. This evidence supports the view that macrophages not only adapt to the TME but also help shape it, providing a favorable environment for tumor survival and therapy resistance [48].

Our results showed that patients in the high-risk group often had a lower tumor mutation burden; their tumors appeared to evade immune surveillance through strong TME-mediated suppression rather than through neoantigen scarcity. Frequent mutations in *TP53* and *PTEN* across both risk groups further underscore these pathways’ prominence in GBM pathogenesis. Drug sensitivity analyses pointed to potential vulnerabilities in the high-risk group, which showed a heightened sensitivity to inhibitors of the IGF-1R and MAPK pathways, such as BMS-536924 [32], SCH772984 [33], and Selumetinib [34]. In contrast, treatments targeting HDAC or inducing apoptosis appeared more effective in the low-risk group. These findings highlight the value of tailoring therapy based on a patient’s molecular and immune profile, and they reinforce the idea that taming TAM-driven immunosuppression may improve the effectiveness of existing treatments.

Emerging evidence suggests that reprogramming tumor-associated macrophages holds significant promise as a therapeutic strategy in glioblastoma, given their central role in shaping the tumor microenvironment [49,50]. However, the substantial heterogeneity among TAM subpopulations poses a challenge, as these cells exhibit diverse functions ranging from promoting to suppressing tumor progression [51]. Several strategies have been proposed to reprogram TAMs toward a proinflammatory, tumoricidal phenotype, including CSF1R inhibition (e.g., PLX3397 and BLZ945), the blockade of IL-10 or TGF-β signaling, the activation of Toll-like receptors (e.g., CpG ODNs), and the inhibition of the STAT3 pathway. These approaches aim to attenuate immunosuppression, enhance antigen presentation, and stimulate cytotoxic T-cell responses. Our findings highlight the importance of targeting high-risk patients, who are closely associated with pathways related to angiogenesis and immunosuppression—key processes that support tumor growth and immune evasion. Disrupting these TAM functions could dismantle critical tumor-sustaining mechanisms. Combining TAM-targeting therapies with immune checkpoint inhibitors could restore T-cell activity, while anti-angiogenic drugs and metabolic disruptors may further enhance therapeutic efficacy by addressing multiple facets of the tumor-supportive niche [52,53]. Such an integrated approach is particularly promising for high-risk glioblastoma patients, who often demonstrate resistance to conventional treatments, paving the way for more effective and durable outcomes [54,55].

Despite these promising results, several important issues must be addressed. First, the risk signature was derived from retrospective datasets and requires validation in prospective clinical studies. Second, although the gene signature demonstrated moderate predictive accuracy, incorporating additional molecular and clinical variables may enhance its prognostic utility. Third, further mechanistic studies are necessary in order to elucidate how *THEMIS2*, *SIGLEC9*, *CSTA*, *LILRB3*, and *MS4A6A* cooperate to influence macrophage biology and immune evasion in GBM. Fourth, while the drug sensitivity analysis suggested potential therapeutic vulnerabilities associated with the gene signature, it was based on transcriptomic predictions derived from in vitro CTRP data, which lack immune cell components. Therefore, these findings should be interpreted as tumor-cell-intrinsic and require rigorous validation in more physiologically relevant systems—such as immune-competent preclinical models or organoid-immune co-culture platforms—before informing therapeutic strategies like TAM-targeting or combination approaches involving immune checkpoint and anti-angiogenic inhibitors. Moreover, any therapeutic strategies informed by these results, including TAM-targeted approaches or combinations of immune checkpoint inhibitors with anti-angiogenic agents, must undergo stringent testing in both preclinical models and prospective clinical trials to ensure safety and efficacy.

Future studies should focus on overcoming the challenges associated with GBM’s heterogeneous TME. Advancements in spatial transcriptomics and multiplex imaging could provide deeper insights into TAM subpopulations and their interactions with other TME components [56,57]. Additionally, preclinical models that better recapitulate GBM’s complex microenvironment are essential for evaluating combination therapies.

## 5. Conclusions

In conclusion, our multi-cohort and single-cell analyses identified a five-gene, immune-related prognostic signature in GBM that aligns with a TAM-centered, immunosuppressive microenvironment and worsened clinical outcomes. By highlighting key molecular, cellular, and microenvironmental underpinnings, our study demonstrates the potential of targeting macrophage-driven immune pathways—possibly in combination with anti-angiogenic and metabolic interventions—to improve therapeutic efficacy in high-risk GBM patients. Future research aimed at refining this prognostic model, elucidating the molecular synergy among its constituent genes, and testing new combination therapies holds promise for enhancing the clinical care of patients with this devastating disease.

## Figures and Tables

**Figure 1 cancers-17-01799-f001:**
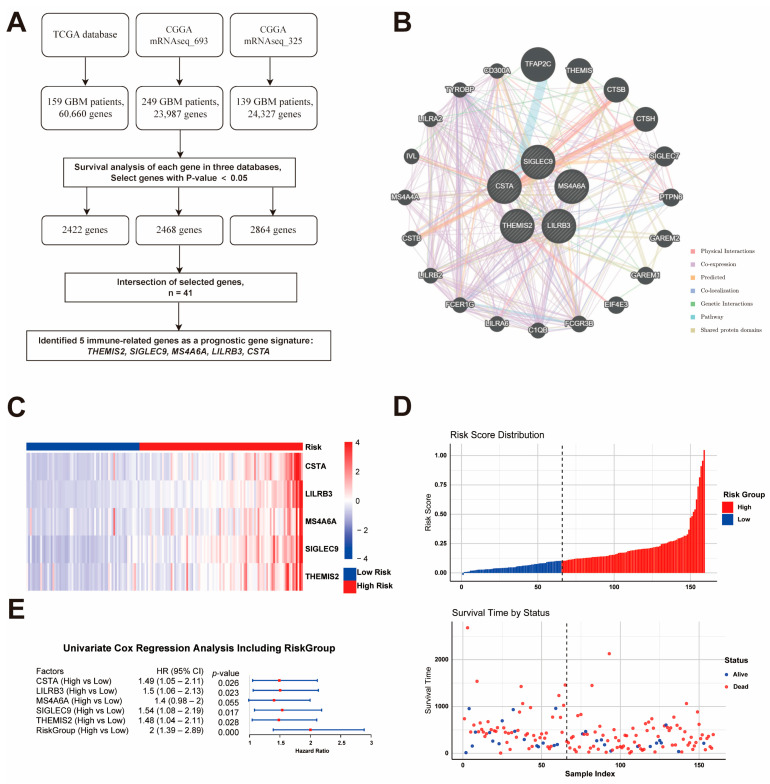
Identification and Validation of the Prognostic Gene Signature. (**A**) Workflow for identifying five immune-related prognostic genes (*THEMIS2*, *SIGLEC9*, *CSTA*, *LILRB3*, and *MS4A6A*) from three GBM datasets. (**B**) Gene interaction network showing functional connections among the five genes and key immune regulators. (**C**) Heatmap of gene expression in high- and low-risk groups based on risk score. (**D**) Risk score distribution and survival status. (**E**) Cox regression analysis confirming the prognostic significance of the gene signature.

**Figure 2 cancers-17-01799-f002:**
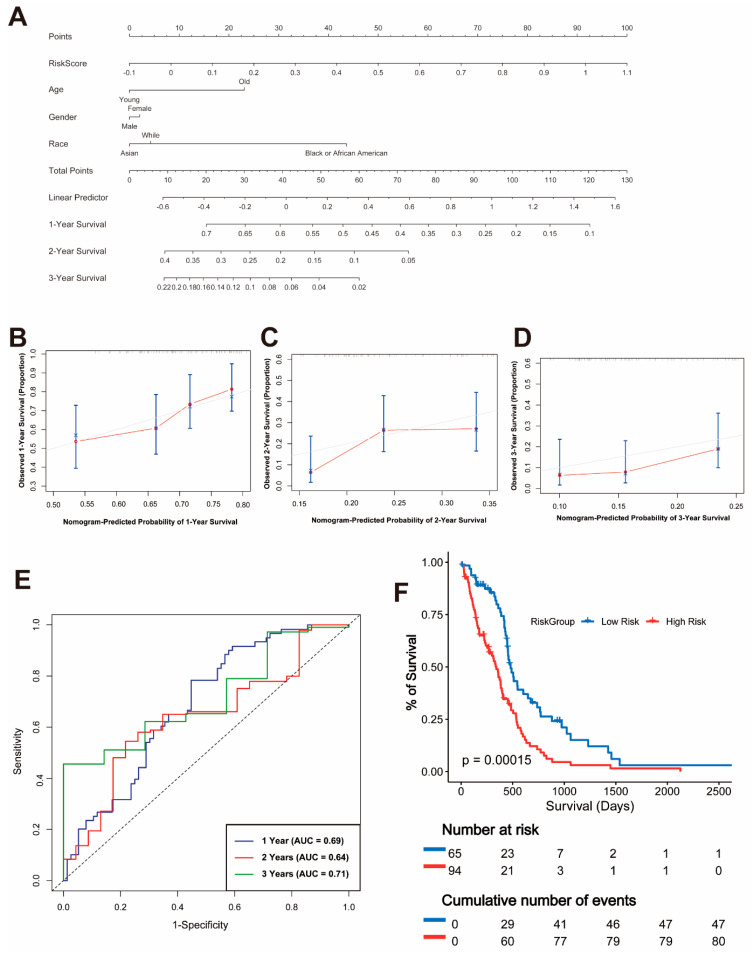
Nomogram and Validation of the Prognostic Model. (**A**) Nomogram integrating risk scores and clinical variables, including age, gender, and race, for predicting 1-year, 2-year, and 3-year overall survival. (**B**–**D**) Calibration plots for 1-year (**B**), 2-year (**C**), and 3-year (**D**) survival probabilities. (**E**) Time-dependent ROC curves for 1-year, 2-year, and 3-year overall survival with corresponding AUC values. (**F**) Kaplan–Meier survival curves for overall survival in high-risk and low-risk groups.

**Figure 3 cancers-17-01799-f003:**
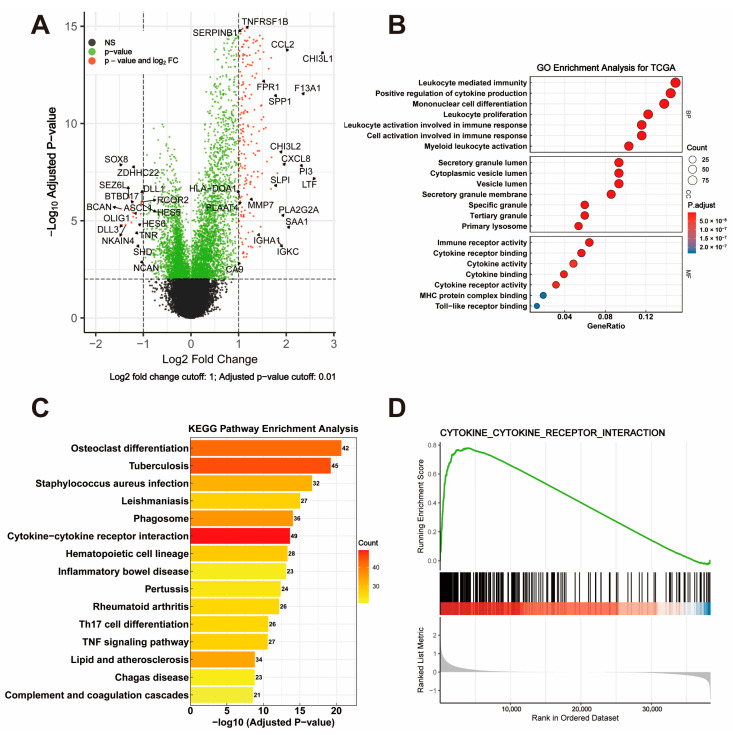
Visualization and Functional Enrichment Analysis of DEGs. (**A**) Volcano plot displaying differentially expressed genes (DEGs) between high-risk and low-risk groups, with criteria set at |log2 fold change| > 1 and adjusted *p*-value < 0.01. (**B**) GO enrichment analysis of DEGs, categorized into biological process (BP), cellular component (CC), and molecular function (MF). (**C**) KEGG pathway enrichment analysis highlighting significant pathways involving DEGs. (**D**) GSEA plot showing enrichment of the cytokine–cytokine receptor interaction pathway.

**Figure 4 cancers-17-01799-f004:**
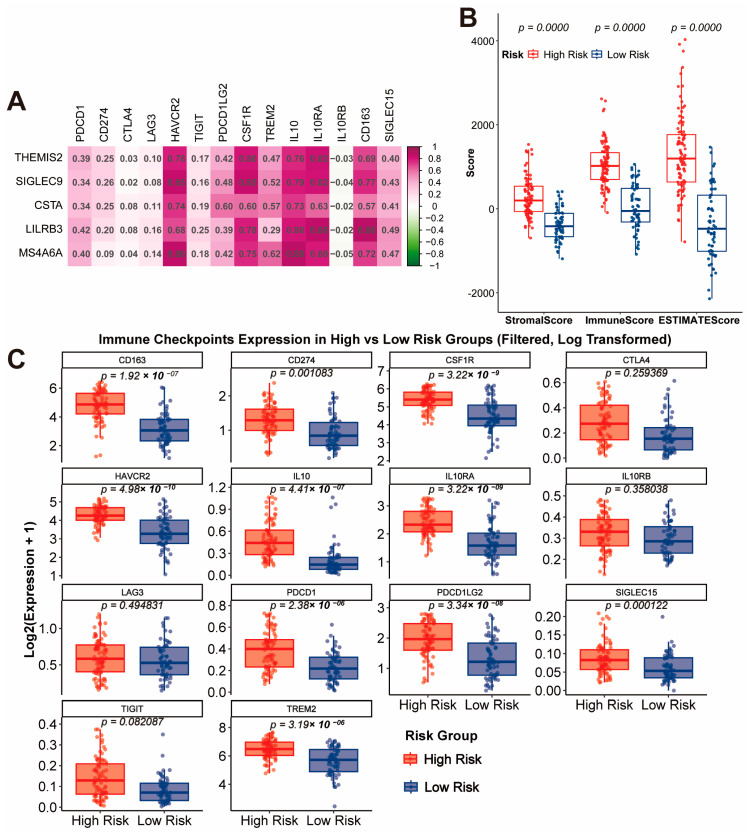
Immune Checkpoint Correlation and Tumor Microenvironment Characteristics. (**A**) Correlation analysis of the gene signature with immune checkpoints. (**B**) Comparison of StromalScore, ImmuneScore, and ESTIMATEScore between high-risk and low-risk groups. (**C**) Expression levels of immune checkpoint molecules in high-risk and low-risk groups.

**Figure 5 cancers-17-01799-f005:**
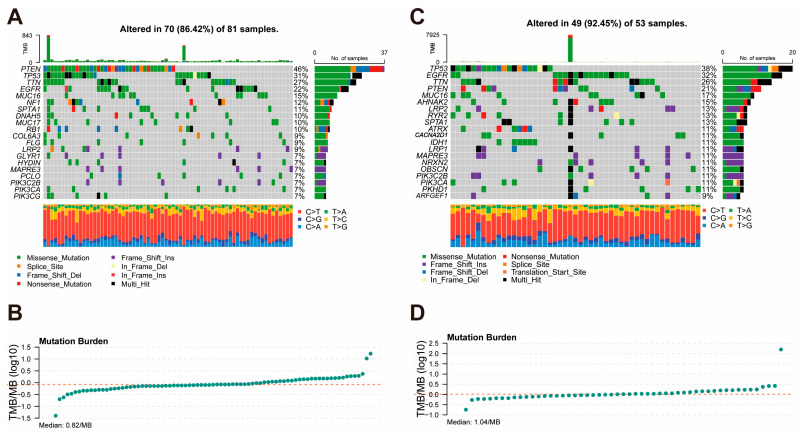
Mutation Landscape and Tumor Mutation Burden in High-Risk and Low-Risk Groups. (**A**) Mutation landscape of the high-risk group, showing the frequency and types of somatic mutations. (**B**) Distribution of tumor mutation burden in the high-risk group. (**C**) Mutation landscape of the low-risk group, displaying the frequency and types of somatic mutations. (**D**) Distribution of tumor mutation burden in the low-risk group.

**Figure 6 cancers-17-01799-f006:**
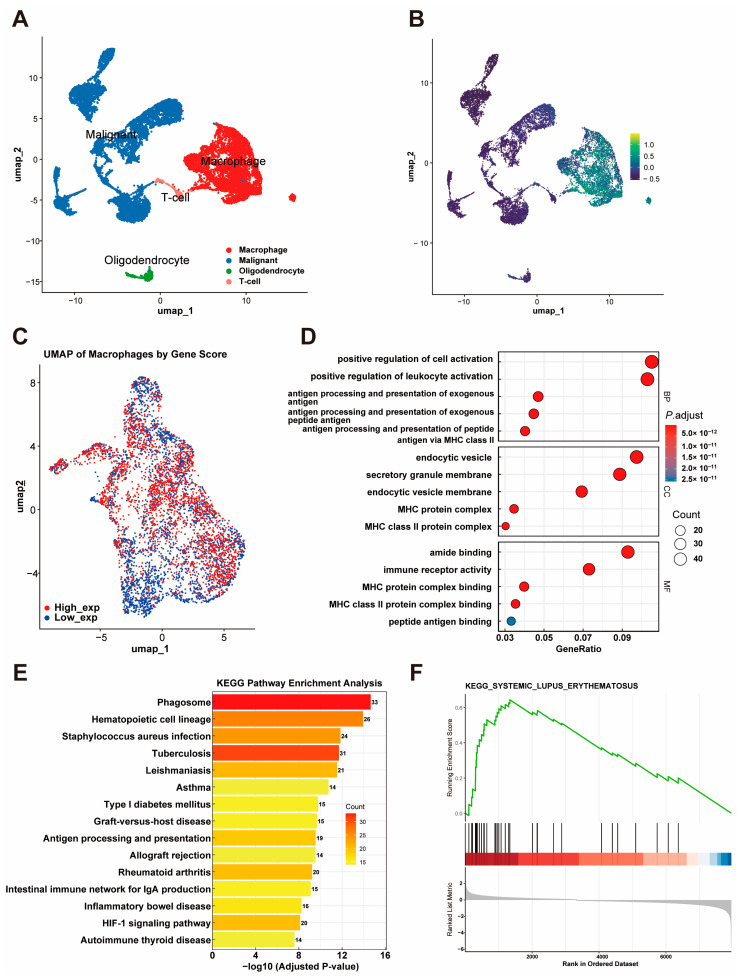
Single-Cell RNA-Seq Analysis of the Gene Signature. (**A**) UMAP plot showing four major cell populations (macrophages, malignant cells, T cells, and oligodendrocytes) classified using the GSM3828673 dataset. (**B**) Expression patterns of THEMIS2, SIGLEC9, CSTA, LILRB3, and MS4A6A across cell populations. (**C**) UMAP visualization of macrophages stratified by gene signature scores, showing distinct high- and low-expression groups. (**D**) GO enrichment analysis of DEGs from macrophages with high and low gene signature expression. (**E**) KEGG pathway enrichment analysis of DEGs. (**F**) GSEA plot showing enrichment of DEGs in the systemic lupus erythematosus pathway.

**Figure 7 cancers-17-01799-f007:**
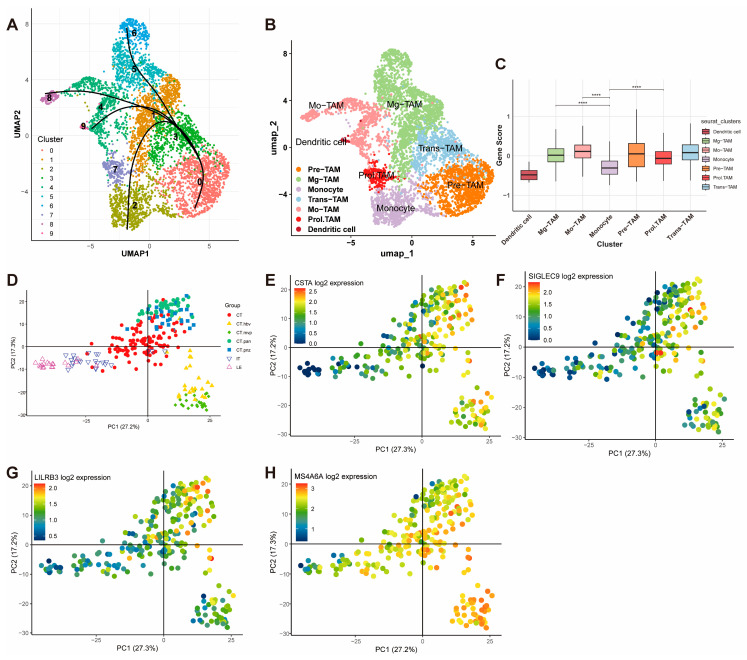
Functional Analysis and Spatial Validation of the Gene Signature in Macrophages. (**A**) Pseudotime trajectory analysis using Slingshot, illustrating the transitional relationships among macrophage subtypes. (**B**) UMAP visualization of macrophage subpopulations classified into seven subtypes: Mg-TAM, Mo-TAM, Prol-TAM, Pre-TAM, Trans-TAM, Monocyte, and Dendritic Cell. (**C**) Module scores of the gene signature across macrophage subtypes (**** *p* < 0.0001). (**D**–**H**) PCA analysis from the Ivy Glioblastoma Atlas Project showing high expression of CSTA, SIGLEC9, LILRB3, and MS4A6A in glioblastoma microenvironmental niches, including CT-HBV, CT-MVP, CT-PAN, and CT-PNZ.

**Figure 8 cancers-17-01799-f008:**
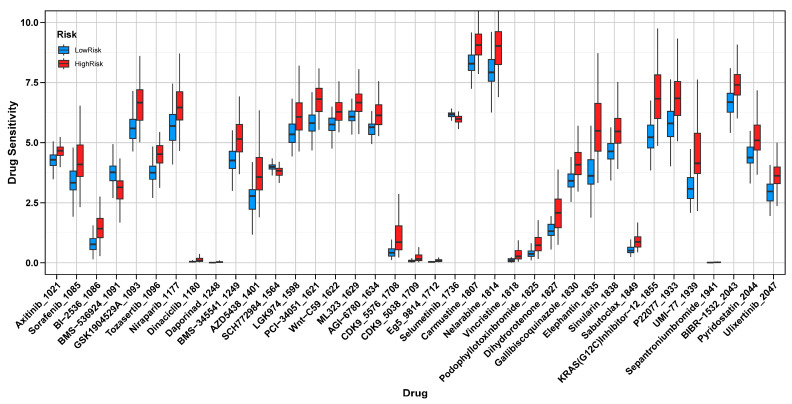
Drug Sensitivity Analysis in High-Risk and Low-Risk Groups. Comparison of IC50 values between high-risk and low-risk groups, showing distinct drug sensitivities.

## Data Availability

The data supporting the findings of this study can be obtained from the corresponding author upon reasonable request. The code used for data analysis is available at https://github.com/cielowq/Prognostic-Immune-Related-Gene-Signature-in-Glioblastoma.git (accessed on 27 March 2025) and archived with DOI: 10.5281/zenodo.15491692.

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
