# Peer review of "Identification of an Immune-Related Gene Signature for Prognostic Prediction in Glioblastoma: Insights from Integrated Bulk and Single-Cell RNA Sequencing"

_cancers, 2025, doi:10.3390/cancers17111799_

Round 1
Reviewer 1 Report
Comments and Suggestions for Authors
This study combined bulk RNA-seq and scRNA-seq data to discover an immune-related gene signature—THEMIS2, SIGLEC9, CSTA, LILRB3, and MS4A6A—strongly correlated with poor survival in glioblastoma (GBM). The study elucidates the immunological landscape of GBM and finds a gene signature with substantial prognostic and therapeutic ramifications. By clarifying the role of proliferating tumor-associated macrophages in tumor growth and identifying their therapeutic vulnerabilities, the authors underscore the necessity of creating targeted therapeutics for these cells.
The manuscript is of high quality, with its aim and topic within the scope of the section of Cancers. Percent match(32%, iThenticate) could be lower, but its value is mostly caused by the properly described methodology section. I appreciate the concise yet informative introduction, with clearly stated aims at its end. Method are rich and allow to fully reproduce the study. The design is OK and the results are properly discussed and concluded. This is how the work should look like!
Minor comments:
Line 140: a reference is needed here, to “rms”
Pages 6-9, why the supplementary figures are presented in the main manuscript?
Line 260, a reference is needed here, to “ggplot2”
The observed AUC = 0.64–0.71 indicates that the model has only moderate predictive power. While the authors admit this in the discussion, it would be worth considering other models (e.g. LASSO, XGBoost) or extending them to include more clinical variables.
I’m a little bit worried about the potential overinterpretation of GSEA/SLE results. While the association with SLE is interesting, no data are presented that clearly confirm the functional similarity of TAMs in GBM and macrophages in SLE. References are needed.
Probably the most important comment is that despite promising observations, no realistic paths for translation of the results into clinical practice (e.g. specific biomarkers, diagnostic tests) are presented. This should be updated.
Reviewer 2 Report
Comments and Suggestions for Authors
In this study, the authors attempt to identify immune-related gene signatures that correlate with GBM with poor survival by using public available datasets from TCGA, CCGA, and GEO databases. The study is well designed, well written, and interesting. However, there is a major flaw in the study that needs to be improved/ corrected, as follows:
- In drug sensitivity analysis, the authors utilize the data from Cancer Therapeutic Response Portal, which compiled 860 characterized cancer cell lines tested with 481 small molecular inhibitor compounds. While this dataset is commonly used in cancer research, the results are mostly based on in vitro cell lines sensitivity assay, which is free from immune-cell response. As this study focuses on immune signature, this dataset may not be applicable to correlate /link the study’s aims and findings. It is recommended that the authors use a more relevant dataset.
In light of these comments, I would like to recommend a major revision for this manuscript, and I look forward to the revised manuscript.
Reviewer 3 Report
Comments and Suggestions for Authors
This work is devoted to identification of an immune-related gene signature for prognostic prediction in glioblastoma. Here authors identified a 5-gene immune-related signature that is significantly associated with worse survival outcomes and increased immune cell infiltration. It was shown that these genes are involved in key immune pathways, including antigen presentation, cytokine signaling, and immune cell activation. This study demonstrates the potential of targeting macrophage-driven immune pathways—possibly in combination with anti-angiogenic and metabolic interventions—to improve therapeutic efficacy in high-risk GBM patients. The manuscript looks like an Article and may be published after minor revision.
Notes:
- Authors should avoid any abbreviations in the Abstract of this manuscript.
- Why did the authors choose only five genes to their study?
- The authors reported in the manuscript that reprogramming tumor-associated macrophages holds significant promise as a therapeutic strategy for glioblastoma. How can this macrophage reprogramming be achieved? Short comment should be added.
- In the article, the authors discuss high-risk and low-risk glioblastoma patients. How do they classify them into one group or another? What indicator is used to classify them into risk groups?
Round 2
Reviewer 2 Report
Comments and Suggestions for Authors
The authors have address all the comments and suggestions from the previous review. The manuscript is now ready to be published. I have no further questions.